# Positive Autoantibody Is Associated with Malignancies in Patients with Idiopathic Interstitial Pneumonias

**DOI:** 10.3390/biomedicines10102469

**Published:** 2022-10-02

**Authors:** Takuma Koga, Masaki Okamoto, Minoru Satoh, Kiminori Fujimoto, Yoshiaki Zaizen, Tomonori Chikasue, Akiko Sumi, Shinjiro Kaieda, Norikazu Matsuo, Goushi Matama, Takashi Nouno, Masaki Tominaga, Kazuhiro Yatera, Hiroaki Ida, Tomoaki Hoshino

**Affiliations:** 1Division of Respirology, Neurology, and Rheumatology, Department of Internal Medicine, Kurume University School of Medicine, Kurume 830-0011, Japan; 2Department of Respirology and Clinical Research Center, National Hospital Organization Kyushu Medical Center, Fukuoka 810-0065, Japan; 3Department of Clinical Nursing, School of Health Sciences, University of Occupational and Environmental Health, Fukuoka 807-8555, Japan; 4Department of Medicine, Kitakyushu Yahata Higashi Hospital, Fukuoka 805-0071, Japan; 5Department of Radiology, Kurume University School of Medicine, Kurume 830-0011, Japan; 6Department of Respiratory Medicine, University of Occupational and Environmental Health, Japan, Fukuoka 807-8555, Japan

**Keywords:** idiopathic interstitial pneumonia, lung cancer, autoantibody, nucleolar antinuclear antibody

## Abstract

Various autoantibodies are associated with clinical outcomes in patients with idiopathic interstitial pneumonias (IIPs). We retrospectively analyzed the association between autoantibodies and malignancies in IIP patients. Comprehensive analyses of autoantibodies were performed using immunoprecipitation and enzyme-linked immunosorbent assays in 193 consecutive IIP patients. Cancer-related factors were analyzed using logistic regression analysis. In total, 22 of 193 patients (11.4%) with IIP had malignant disease. In univariate analysis, positivity for any autoantibody (odds ratio (OR), 3.1; 95% confidence interval (CI), 1.2–7.7; *p* = 0.017) and antinuclear antibody titer ≥1:320 (OR, 3.4; CI, 1.2–9.8; *p* = 0.024) were significantly associated with malignancies. Positive anti-aminoacyl tRNA synthetase (ARS) (OR, 3.7; CI, 0.88–15.5; *p* = 0.074) and anti-Ro52 antibody (OR, 3.2; CI, 0.93–11.2; *p* = 0.065) tended to be associated with malignancies. In multivariate analysis, independent risk factors were male sex (OR, 3.7; CI, 1.0–13.5; *p* = 0.029) and positivity for any autoantibody (OR, 3.9; CI, 1.5–10.1; *p* = 0.004) in model 1, and male sex (OR, 3.9; CI, 1.0–15.3; *p* = 0.049), antinuclear antibody titer ≥1:320 (OR, 4.2; CI, 1.4–13.3; *p* = 0.013), and positivity for anti-ARS antibody (OR, 6.5; CI, 1.2–34.1; *p* = 0.026) in model 2. Positivity for any autoantibody, antinuclear and anti-ARS antibodies, and male sex were independent risk factors for malignancies in IIP patients. Testing autoantibodies in IIP patients might help the early diagnosis of malignancies.

## 1. Introduction

Idiopathic interstitial pneumonias (IIPs) are diffuse inflammatory and/or fibrotic lung diseases that exclude interstitial lung diseases (ILDs) with known causes such as connective tissue diseases (CTDs) [1,2,3]. Evaluation of chest computed tomography (CT) supports the early diagnosis of ILDs [4,5]. Recent studies have identified the key role of crosstalk among dysregulated epithelial cells, mesenchymal cells, immune cells, endothelial cells, genetic mutations and environmental factors (e.g., smoking) in idiopathic pulmonary fibrosis, a subtype of IIP [6]. Patients with IIPs occasionally have clinical features suggesting the presence of autoimmune diseases, including positive autoantibodies, without meeting the diagnostic criteria for any CTD. For example, disease-specific symptoms are often lacking in patients with early or mild systemic sclerosis [5]. Interstitial pneumonia with autoimmune features (IPAF) is a term that represents a subset of patients with this phenotype [7]. Some autoantibodies, in particular, myositis-specific autoantibodies such as anti-aminoacyl tRNA-synthetase (ARS) and anti-melanoma differentiation-associated gene 5 (MDA-5) antibodies, were reported to be associated with the unique clinical features of polymyositis/dermatomyositis (PM/DM). Anti-ARS antibody is often detected in patients with ILD and is associated with more favorable responses to initial therapy, as well as relapses and chronic deterioration [8,9]. The presence of anti-MDA-5 antibody is associated with the development of fatal rapidly progressive ILD [10,11]. In patients with IIP, 5.9–10.7% of cases are positive for anti-ARS antibodies, but anti-MDA-5 antibody-positive cases are rare [10,12,13,14,15]. Anti-ARS antibody-positive IIP patients are more frequently treated with corticosteroids and/or immunosuppressants, have a younger median age, joint or skin involvement, or higher KL-6 or lower forced vital capacity than antibody-negative cases [12,13,14,15]. Anti-Ro52/tripartite motif-containing (TRIM) 21 antibodies (anti-Ro52) are commonly detected in Sjögren’s syndrome, PM/DM, and systemic sclerosis [16]. Anti-Ro52 antibodies often coexist with anti-ARS antibodies (particularly anti-Jo-1 and anti-PL-7) in patients with PM/DM [17,18]. The presence of anti-Ro52 antibodies was associated with a higher prevalence of ILD [19]. Chen et al. reported that 5 of 20 patients with PM/DM-ILD with anti-Ro-52 antibodies developed rapidly progressive ILD and responded well to therapy, with a good prognosis [19]. Moreover, a two-center retrospective study of 288 consecutive patients with IIP showed that anti-Ro52 (20/288; 6.9%), anti-ARS (18/288; 6.3%), and anti-Ro60 (16/288; 5.6%) were the most common autoantibodies detected, and that anti-ARS antibodies were present in 8 (40%) of 20 IIP patients with anti-Ro52 antibodies [20]. The criteria for IPAF were significantly better fulfilled by patients with anti-Ro52 antibodies than those without [20]. The presence of autoantibodies is a potential biomarker for predicting the prognosis and complications in patients with CTD-ILD, as well as IIP, as shown in several studies [12,13,14,15,20]. It was also shown that patients with IIP often develop malignant disease. A previous large cohort study found that the overall incidence of cancer in patients with idiopathic pulmonary fibrosis (IPF) was 37.3 cases per 1000 person years, and that their overall cancer incidence risk was significantly higher than that of the general population (relative risk, 1.51; 95% confidence interval [CI], 1.20–1.90). Lung cancer had the highest risk, followed by lymphoma, skin cancer, uterine cervical cancer, multiple myeloma, thyroid cancer, leukemia, pancreatic cancer, liver cancer, and prostate cancer. Moreover, adjusting for the effects of smoking and other cancer-associated covariates had little effect on the hazard ratio of overall and specific cancers. [21]. Malignant disease is one of the major risk factors responsible for the reduced survival of patients with IIP, because surgery or drug therapy for malignant disease can trigger lethal respiratory failure caused by acute exacerbation of IIP [22,23]. The early diagnosis and treatment of malignant disease in patients with IIP is therefore an important issue in clinical practice.

Autoantibodies are more frequently found in patients with various cancers compared with the healthy population. Previous reports showed that autoantibodies can act as potential diagnostic biomarkers for malignant disease [24,25,26,27,28,29,30,31]. An increase in the prevalence of autoantibodies among patients with malignant disease was reported to be associated with the evolution of immune responses against cancer-associated proteins related to mutations in cancer tissues, epitope spreading, and other mechanisms [32,33]. However, the association between the presence of autoantibodies and malignancies in patients with IIP has not been sufficiently elucidated.

We aimed to clarify the association between the presence of autoantibodies and malignancies in patients with IIP, on the basis of these previous studies. To the best of our knowledge, this represents the first study to analyze the association between autoantibodies and malignancies in IIP.

## 2. Materials and Methods

### 2.1. Patients

We retrospectively analyzed the medical records and sera from 193 consecutive IIP patients at the time of their initial diagnosis at our hospital between 2012 and 2021. All patients with IIP were diagnosed retrospectively and classified by multidisciplinary discussion (MDD) diagnosis using medical records, chest X-ray, and high-resolution CT (HRCT) based on the global guidelines for IIP [1,3]. MDD diagnoses were performed by two respiratory physicians and three chest radiologists and/or one chest pathologist specializing in diffuse lung disease. The classification criteria of IPAF were based on the 2015 European Respiratory Society/American Thoracic Society Task Force research statement [7]. Acute exacerbation (AE) of ILD was defined based on a previous report [34]. Patients diagnosed with secondary ILDs, such as hypersensitivity pneumonia, CTD, or pneumoconiosis, were excluded from this study. In this study, both former and current smokers were defined as smokers (Table 1).

### 2.2. Analyses of Serum Autoantibodies

Serum samples were obtained from patients with IIP at their first visit and stored at −80°C. Autoantibodies were measured by immunoprecipitation of ^35^S-methionine/cysteine radiolabeled K562 cell extracts with IgG purified from 8 μL of human serum. The immunoprecipitated proteins were electrophoresed by sodium dodecyl sulfate-polyacrylamide gel electrophoresis (SDS-PAGE), as described previously [25]. In summary, cells were labeled with ^35^S-methionine and cysteine, lysed in 0.5 M NaCl, 2 mM ethylenediaminetetraacetic acid (EDTA), 50 mM Tris (pH 7.5), 0.3% octylphenyl polyethylene glycol (IGEPAL CA-630) buffer containing 0.5 mM phenylmethylsulfonyl fluoride, and 0.3 trypsin inhibitory units (TIU)/mL of aprotinin, and then immunoprecipitated using protein-A-Sepharose beads coated with IgG. The immunoprecipitates were washed with 0.5 M NaCl-NET/IGEPAL CA-630 and analyzed by SDS-PAGE and autoradiography. The specificity of the autoantibodies was confirmed using human reference sera [25]. Antibodies to Ro52, histidyl-tRNA synthetase (Jo-1), centromere protein A and B (CENP-A and CENP-B), and MDA-5 were measured by enzyme-linked immunosorbent assays, as described previously [25]. All recombinant proteins were purchased from Diarect (Freiburg, Germany). Briefly, 96-well microtiter plates (Immobilizer Amino; Nunc Naperville, IL, USA) were coated with 0.5 μg/mL of recombinant protein and blocked with 0.5% bovine serum albumin (BSA)-NET/IGEPAL CA-630 for 1 h at room temperature. Patients’ sera (1:250) and alkaline phosphatase-conjugated goat anti-human IgG (1:1000; γ-chain specific; Jackson Immunoresearch, West Grove, PA, USA) diluted in 0.5% BSA-NET/IGEPAL CA-630 were used as the sample and secondary antibodies, respectively. A standard curve was generated using serial 1:5 dilutions of a high-titer prototype serum. The optical density of the samples measured at 405 nm was converted into units based on the standard curve. Anti-nuclear antibodies (ANA), anti-cyclic citrullinated peptide (CCP) antibodies, and rheumatoid factor (RF) were tested commercially, and positivity was defined in accordance with the classification criteria of IPAF [7]: ANA titer ≥ 1:320, diffuse, speckled, homogeneous patterns; ANA any titer, nucleolar or discrete speckled patterns; RF ≥ 2× upper limit of normal.

### 2.3. Statistical Analyses

Data are expressed as the median (25th–75th percentiles of the interquartile range). We analyzed differences in clinical parameters between two groups using Wilcoxon’s rank-sum or Fisher’s exact test. Correlations between the two parameters were evaluated using Spearman’s rank correlation coefficient. For logistic regression analysis, we detected variables significantly (*p* < 0.05) associated with malignancies by univariate analyses. Variables that were positive in ≥5 of 193 patients were analyzed by Cox proportional hazards regression analysis. Variables with *p* < 0.10 in the univariate analyses, as well as age at the time of diagnosis with IIP, male sex, and smoking status, as common risk factors of malignant disease, were analyzed by multivariate analysis using the backward elimination method. A *p*-value < 0.05 was considered statistically significant. Statistical analysis was performed using JMP 16.0 (SAS Institute, Cary, NC, USA).

## 3. Results

### 3.1. Patient Characteristics and Clinical Outcomes

The patients’ characteristics and clinical outcomes are shown in Table 1. The study cohort consisted of 193 patients (141 males; median age, 68.0 years at the time of diagnosis with ILD), including 99 (51.3%) with IPF, 7 (3.6%) with non-specific interstitial pneumonia (NSIP), 1 (0.52%) with cryptogenic organizing pneumonia, and 86 (44.2%) with unclassifiable IIP. In total, 73 of the 193 patients (37.8%) died during the observation period. Mortality was higher among the 22 patients with malignant disease than among the remaining 171 patients (68.2% vs. 33.9%, *p* = 0.004). In total, 15 patients with malignant disease died during the observation period due to malignant disease in 9 (60.0%), AE in 3 (20.0%), ILD in 1 (6.7%), infectious disease in 1 (6.7%), and other causes in 1 (6.7%). There were no significant differences between groups in age, levels of serum biomarkers, pulmonary function, HRCT features at the time of diagnosis with IIP, sex, smoking status, ILD subtype, prevalence of diagnosis with IPAF, and developing AE and CTD.

### 3.2. Analyses of Autoantibodies

The results of comprehensive analyses of autoantibodies are presented in Table 2. In total, 76 of 193 patients (39.4%) had at least one of the listed autoantibodies. Positive number and prevalence of individual autoantibody subtypes present in >5% of all IIP patients were as follows: anti-CCP antibody in 12 patients (6.2%), anti-double stranded DNA antibody in 14 (7.3%), anti-Ro52 antibody in 15 (7.8%), anti-ARS antibody in 10 (5.2%), ANA with a titer ≥1:320 in 23 (11.9%), ANA showing a nucleolar pattern in 12 (6.2%), and >2× upper limit of RF in 28 (14.5%). Autoantibodies against La, ribonucleoprotein (RNP), Sm, TIF-1γ, TIF-1β, and Ku were not detected in any patients with IIP. Among the 10 patients with anti-ARS antibody, the antibody subtypes included PL-7, PL-12, EJ, and KS in 2 patients each, and Jo-1 and OJ in 1 patient each. The coexistence of anti-Ro52 and anti-ARS antibodies was seen in two patients (20%), and the antibody subtypes were EJ in both patients. One of these patients had breast cancer (Case 17 in Table 3) and the other did not have malignant disease.

The prevalence of having any of the listed autoantibodies (63.6% vs. 36.4%, *p* = 0.019) and an ANA titer ≥ 1:320 (27.3% vs. 9.9%, *p* = 0.030) was significantly higher in patients with malignant disease than in the remaining patients. Similarly, the prevalence of anti-Ro52 antibodies (18.2% vs. 6.4%, *p* = 0.074), anti-ARS antibodies (13.6% vs. 4.1%, *p* = 0.091), and anti-Ki/SL antibodies (12.5% vs. 0.54%, *p* = 0.081) in patients who developed malignant disease tended to be higher than in the remaining patients. There was no significant difference in the prevalence of other autoantibodies between patients with and without malignant disease.

### 3.3. Patient Characteristics and Clinical Outcomes of IIP Patients with Malignant Disease

The characteristics and clinical outcomes of the 22 patients with malignant disease are presented in Table 3. These patients included 19 males (86.3%) and 17 smokers (77.3%). ILD subtypes consisted of IPF in 13 patients (59.1%), unclassifiable IIP in 8 (36.4%), and NSIP in 1 (4.5%). Cancer subtypes in these patients consisted of primary lung cancer in 9 patients (40.9%), gastric cancer in 4 (18.2%), rectal cancer in 3 (13.6%), esophageal, pharyngeal, breast, ovarian, bladder, prostate, liver, and skin cancer in 1 (4.5%) each. In 5 of the 22 patients (22.7%), malignant disease was diagnosed a median of 1461 (168.0–1673.5) days before IIP. In the remaining 17 cases (77.3%), IIP was diagnosed a median of 478.0 (91.0–809.5) days before the malignant disease. Treatments for malignant diseases included surgery in 13 patients (59.1%), palliative care in 6 (27.3%), drug therapy in 2 (9.1%), and transcatheter arterial chemoembolization with radiofrequency ablation in 1 (4.5%). Fifteen of the twenty-two patients (68.2%) died, and the overall survival was 1294.0 (810.8–1715.3) days (Table 1).

### 3.4. Univariate and Multivariate Logistic Regression Analyses

The results of univariate logistic regression analysis are shown in Table 4. Positivity for any autoantibodies (odds ratio (OR), 3.1; 95% CI, 1.2–7.7; *p* = 0.017) and ANA titer ≥ 1: 320 (OR, 3.4; 95% CI, 1.2–9.8; *p* = 0.024) were significantly associated with malignant disease. Anti-ARS antibodies (OR, 3.7; 95% CI, 0.88–15.5; *p* = 0.074) and anti-Ro52 antibodies (OR, 3.2; 95% CI, 0.93–11.2; *p* = 0.065) tended to be associated with malignant disease.

The results of multivariate analyses using the backward elimination method are shown in Table 5. Positivity for any autoantibody and individual autoantibodies (i.e., ANA titer ≥ 1:320, and positive for anti-ARS and anti-Ro52 antibodies) was analyzed in separate models because these were considered confounding factors. Variables in model 1 included positivity for any autoantibody, age at the time of diagnosis with IIP, male sex, and smoking status. Similarly, variables in model 2 included ANA titer ≥ 1:320, positivity for anti-ARS and anti-Ro52 antibodies, age at the time of diagnosis with IIP, male sex, and smoking status. In model 1, independent risk factors for malignant disease were male sex (OR, 3.7; 95% CI, 1.0–13.5; *p* = 0.029) and positivity for any autoantibody (OR, 3.9; 95% CI, 1.5–10.1; *p* = 0.004). Similarly, the independent risk factors in model 2 were male sex (OR, 3.9; 95% CI, 1.0–15.3; *p* = 0.049), ANA titer ≥1:320 (OR, 4.2; 95% CI, 1.4–13.3; *p* = 0.013), and anti-ARS antibodies (OR, 6.5; 95% CI, 1.2–34.1; *p* = 0.026).

## 4. Discussion

In the present study, we demonstrated that any autoantibodies positively and male sex were independent risk factors for malignancies in patients with IIPs. Positivity for ANA and anti-ARS antibodies in particular had clinical significance among the autoantibody subtypes. The presence of anti-Ro52 antibodies tended to be associated with malignant disease by univariate analysis but was not an independent risk factor in multivariate analysis. These results may be related to the evolution of immune responses against cancer-associated proteins in patients with IIP [32,33].

Previous studies reported that the presence of ANA was associated with various cancers [24,25]. Nishihara et al. showed that the prevalence of ANA was higher in patients with breast cancer than in those with benign tumors among 91 patients with a breast mass assessed by biopsy histology [24]. Other reports suggested that the presence of ANA showing a nucleolar pattern by immunofluorescence was more strongly associated with malignant disease compared with other staining patterns [25]. In the present study, the presence of nucleolar-pattern ANA was not significantly associated with malignant disease. A meta-analysis of 27 studies including 3487 patients with idiopathic inflammatory myositis showed that anti-ARS autoantibodies were identified in 13% of patients with cancer-associated myositis [26]. Shibata et al. reported a case of myositis with anti-OJ antibodies who achieved complete remission and the disappearance of anti-OJ antibodies after lung cancer resection without immunosuppressive therapy, and suggested that anti-ARS antibodies might be linked to the anti-tumor response [27]. A large cohort study of 490 patients with various malignant diseases showed that anti-Ro52 antibodies were significantly more prevalent in patients with ovarian cancer (30%) compared with patients with six other malignant diseases (median 8.1%, range 5.9–15.8%) [28]. A retrospective study of 89 consecutive anti-Jo-1-positive patients with anti-synthetase syndrome showed that cancers were more frequent in 36 patients (40.4%) with coexisting anti-Ro52 antibodies than in the 53 patients without anti-Ro52 antibodies (19.4% vs. 5.7%, *p* = 0.02) [29]. In the present study, one of two patients with anti-ARS and anti-Ro52 antibodies had malignant disease. Previous reports showed that the presence of TIF-1γ antibodies was associated with malignant disease, but this antibody was rarely detected in DM patients with ILD [30,31]. In the present study, none of 193 patients with IIP had anti-TIF-1γ antibodies. A systematic review of 6 studies including 312 patients with DM showed that the sensitivity and specificity of anti-TIF-1γ antibodies for diagnosing cancer-associated DM in 66 patients (21.2%) were 78% (95% CI 45–94%) and 89% (95% CI 82–93%), respectively [30]. However, Kaji et al. reported that none of 7 patients with anti-TIF-1γ antibodies had ILD, and none of 29 with DM-ILD had anti-TIF-1γ antibodies among 52 DM patients [31]. None of 193 patients with IIP in the present study had anti-TIF-1γ antibodies.

Previous reports suggested that the risk of lung cancer was higher in patients with IPF than in those without IPF [21,35,36]. Lee et al. showed that the cancer incidence in patients with IPF was 29.0 cases per 1000 person years, which was significantly higher in than that in the non-IPF group (hazard ratio, 2.09; 95% CI, 1.96–2.16) [21]. The reason for a diagnosis of IPF not being significantly associated with malignant disease in the present study is unclear. In this study, 68 cases (44.3%) had an unclassifiable subtype of IIP. The present study also used a retrospective MDD diagnosis based on historical and limited data from medical records in some patients. It is therefore possible that some patients with IPF were classified as unclassifiable IIP, which might have influenced the results of the analyses. The association between a diagnosis of IPF and malignant disease should thus be analyzed further in future studies.

This study had some limitations. First, the sample size was small, mainly because it was a single-center study. Second, the presence of any autoantibodies, or individual autoantibody subtypes including ANA and anti-ARS antibodies, did not show high sensitivity and specificity for an association with malignant disease. However, the development of malignant disease is influenced by a combination of risk factors, such as smoking, chronic tissue injury, viral infection, environmental exposure, and other genetic or epigenetic factors [21]. Evaluating a combination of autoantibodies with various cancer risk factors may thus help improve the accuracy of predicting malignant disease.

## 5. Conclusions

In the present study, the presence of any autoantibodies, ANA and anti-ARS antibodies, and male gender were independent risk factors for cancer in patients with IIPs. The presence of anti-Ro52 antibodies tended to be associated with malignant disease in univariate analysis. Testing for autoantibodies might help the early diagnosis of malignant disease in patients with IIPs.

## Figures and Tables

**Table 1 biomedicines-10-02469-t001:** Characteristics of patients and outcomes.

		Having	Not Having	
	All Cases	Malignant Disease	Malignant Disease	*p*-Value
**N**	193	22 (11.4%)	171 (88.6%)	
**Age (years)**	68.0 (63.0–74.0)	70.5 (64.8–72.3)	67.0 (63.0–74.0)	0.048/0.506
**Male gender**	141 (73.1%)	19 (86.3%)	122 (71.4%)	0.108/0.201
**Smoker**	126 (65.3%)	17 (77.3%)	109 (64.5%)	0.086/0.339
**ILD subtype**				
IPF	99 (51.3%)	13 (59.1%)	86 (50.3%)	0.055/0.586
NSIP	7 (3.6%)	1 (4.6%)	6 (3.5%)	
COP	1 (0.5%)	0	1 (0.58%)	
Unclassifiable IIP	86 (44.3%)	8 (33.4%)	78 (45.6%)	
**HRCT feature**				
UIP	55 (28.5%)	7 (31.8%)	48 (28.1%)	0.026/0.804
NSIP	21 (10.9%)	2 (9.1%)	19 (11.1%)	−0.021/1.000
NSIP with OP	9 (4.7%)	2 (9.1%)	7 (4.1%)	0.075/0.273
Unclassifiable pattern	59 (30.6%)	6 (27.3%)	53 (31.0%)	−0.026/0.810
**Meeting IPAF criteria**	31 (16.1%)	6 (27.3%)	25 (14.6%)	0.110/0.132
**Serum biomarkers**				
KL-6 (IU/mL)	852.5 (523.3–1302.5)	974.5 (652.5–1284.8)	831.5 (514.8–1323.0)	0.043/0.560
CRP (mg/dL)	0.19 (0.070–0.55)	0.26 (0.060–0.92)	0.18 (0.070–0.54)	0.029/0.698
**Pulmonary function**				
FVC (%)	82.9 (65.3–97.5)	84.0 (70.9–99.5)	82.9 (63.8–97.1)	0.075/0.317
D_LCO_ (%)	71.9 (57.3–89.9)	72.2 (49.2–89.1)	71.3 (58.0–90.0)	−0.020/0.812
**Overall survival (days)**	1366.0 (742.0–2061.5)	1294.0 (810.8–1715.3)	1376.0 (708.0–2112.0)	−0.030/0.678
**Mortality**	73 (37.8%)	15 (68.2%)	58 (33.9%)	0.225/0.004 *
**Cause of death**				
ILD/AE/infection/Malignancies/Others	29 (39.7%)/20 (27.4%)/11 (15.1%)/9 (12.3%)/4 (5.5%)	1 (6.7%)/3 (20.0%)/1 (6.7%)/9 (60.0%)/1 (6.7%)	28 (48.3%)/17 (29.3%)/10 (17.2%)/0/3 (5.2%)	<0.0001 *
**Developing AE**	42 (21.8%)	5 (22.7%)	37 (21.6%)	0.008/1.000
**Developing CTD**	6 (3.1%)	2 (9.1%)	4 (2.3%)	0.124/0.140

Data were expressed as the median (25th to 75th percentiles of the interquartile range [IQR]), unless otherwise stated. IPF: idiopathic pulmonary fibrosis; HRCT: high-resolution computed tomography; IIP: idiopathic interstitial pneumonia; ILD: interstitial lung disease; NSIP: non-specific interstitial pneumonia; COP: cryptogenic organizing pneumonia; UIP: usual interstitial pneumonia; IPAF: interstitial pneumonia with autoimmune features; FVC: forced vital capacity; D_LCO_: diffusing capacity of the lung for carbon monoxide; AE: acute exacerbation; CTD: connective tissue disease. Age, ILD subtype, HRCT features, serum biomarker level, and pulmonary function were evaluated at the time of diagnosis with IIP. * A *p*-value < 0.05 represented statistical significance by Wilcoxon’s rank-sum or Fisher’s exact test.

**Table 2 biomedicines-10-02469-t002:** Prevalence of autoantibodies.

		Having	Not Having	
	All Cases	Malignant Disease	Malignant Disease	*p*-Value
**N**	193	22 (11.4%)	171 (88.6%)	
**Positive for any autoantibody**	76 (39.4%)	14 (63.6%)	62 (36.3%)	0.178/0.019 *
**Autoantibodies included in the criteria of IPAF**				
Antinuclear antibody				
ANA titer ≧ 320	23 (11.9%)	6 (27.3%)	17 (9.9%)	−0.170/0.030 *
Nucleolar pattern	12 (6.2%)	3 (13.6%)	9 (5.3%)	0.110/0.143
Discrete speckled pattern	6 (3.1%)	1 (4.6%)	5 (2.9%)	0.030/0.521
RF more than upper limit × 2	28 (14.5%)	3 (13.6%)	25 (14.6%)	−0.009/1.000
Anti-CCP antibody	12 (6.4%)	1 (4.6%)	11 (6.6%)	−0.027/1.000
Anti-double stranded DNA antibody	14 (7.5%)	3 (14.3%)	11 (6.6%)	0.092/0.197
Anti-Ro52 antibody	15 (7.8%)	4 (18.2%)	11 (6.4%)	0.140/0.074
Anti-Ro60 antibody	3 (1.6%)	0	3 (1.8%)	−0.045/1.000
Anti-La antibody	0	0	0	
Anti-U1RNP antibody	0	0	0	
Anti-Sm antibody	0	0	0	
Anti-topoisomerase I antibody	1 (0.52%)	0	1 (0.58%)	−0.026/1.000
Anti-ARS antibody	10 (5.2%)	3 (13.6%)	7 (4.1%)	0.137/0.091
Anti-MDA-5 antibody	1 (0.52%)	0	1 (0.58%)	−0.026/1.000
**Autoantibodies not included in criteria of IPAF**				
Anti-CENP-A antibody	5 (2.6%)	1 (4.6%)	4 (2.3%)	0.044/0.457
Anti-CENP-B antibody	7 (3.6%)	1 (4.6%)	6 (3.5%)	0.018/0.578
Anti-RNAP I/III antibody	2 (1.0%)	1 (4.6%)	1 (0.58%)	0.124/0.255
Anti-TIF-1γ antibody	0	0	0	
Anti-TIF-1β antibody	0	0	0	
Anti-Ku antibody	0	0	0	
Anti-Ki/SL antibody	2 (1.0%)	1 (12.5%)	1 (0.54%)	0.124/0.081
Anti-Su/Ago2 antibody	4 (2.1%)	1 (4.6%)	3 (1.8%)	0.062/0.386

RF: rheumatoid factor; CCP: cyclic citrullinated peptide; RNP: ribonucleoprotein; ARS: aminoacyl tRNA synthetase; CENP: centromere protein; MDA-5: melanoma differentiation-associated gene 5; TIF-1: transcriptional intermediary factor 1. * A *p*-value < 0.05 represented statistical significance by Wilcoxon’s rank-sum or Fisher’s exact test.

**Table 3 biomedicines-10-02469-t003:** Clinical characteristics of patients with IIPs who developed malignant disease.

Case	Cancer Type	ILD Subtype	Gender	Age	Smoking Status	Autoantibody	Therapy for Cancer	Outcome Cause of Death
**1**	**Lung**	IPF	Male	71	Smoker	ds-DNA	Palliative	Day 1674, died
**2**	**Lung**	IPF	Male	55	Smoker	None	Drug	Day 1640, died
**3**	**Lung**	UC-IIP	Male	72	Smoker	Nucleolar ANA titer ≧ 1:320	Palliative	Day 863, alive
**4**	**Lung**	UC-IIP	Male	72	Smoker	Speckled ANA titer ≧ 1:320, Ro52, Ki/SL	Palliative	Day 836, died
**5**	**Lung**	IPF	Male	75	Smoker	None	Palliative	Day 378, died
**6**	**Lung**	IPF	Male	70	Smoker	None	Surgery	Day 322, died
**7**	**Lung**	IPF	Male	73	Smoker	None	Drug	Day 450, died
**8**	**Lung**	IPF	Male	71	Smoker	RF, Nucleolar ANA	Palliative	Day 939, died
**9**	**Lung Esophagus Pharynx**	IPF	Male	61	Smoker	None	Surgery	Day 2711, alive
**10**	**Gastric**	IPF	Male	80	Smoker	CCP	Palliative	Day 1070, died
**11**	**Gastric**	IPF	Male	70	Non-smoker	None	Surgery	Day 1836, died
**12**	**Gastric**	UC-IIP	Male	72	Smoker	None	Surgery	Day 1957, alive
**13**	**Gastric**	UC-IIP	Male	70	Smoker	Discrete speckled ANA titer ≧ 1:320, CENP-A, CENP-B	Surgery	Day 675, died
**14**	**Rectal**	UC-IIP	Female	60	Non-smoker	ARS (PL-7)	Surgery	Day 1675, alive
**15**	**Rectal**	IPF	Male	72	Smoker	Speckled and homogeneous ANA titer ≧ 1:320, RF, dsDNA	Surgery	Day 1667, alive
**16**	**Rectal**	IPF	Male	55	Smoker	None	Surgery	Day 1222, alive
**17**	**Breast**	UC-IIP	Female	66	Non-smoker	ARS (EJ), Ro52	Surgery	Day 1366, died
**18**	**Breast Ovarium**	UC-IIP	Female	68	Non-smoker	Speckled ANA titer ≧ 1:320, Ro52, Su/Ago	Surgery	Day 1411, died
**19**	**Bladder**	UC-IIP	Male	68	Non-smoker	RF	Surgery	Day 936, died
**20**	**Prostate**	NSIP	Male	51	Smoker	ARS (KS)	Surgery	Day 7775, died
**21**	**Liver**	IPF	Male	79	Smoker	RF, dsDNA	RFA, TACE	Day 656, died
**22**	**Skin**	IPF	Male	82	Smoker	Nucleolar ANA titer > 1:320, Ro52, RNAP I/III	Surgery	Day 863, died

ILD: interstitial lung disease; HRCT: high-resolution computed tomography; IPF: idiopathic pulmonary fibrosis; UC-IIP: unclassifiable idiopathic interstitial pneumonia; NSIP: non-specific interstitial pneumonia; TACE: transcatheter arterial chemoembolization; RFA: radio-frequency ablation; ANA: antinuclear antibody; RF: rheumatoid factor; CCP: cyclic citrullinated peptide; ARS: aminoacyl tRNA synthetase; CENP: centromere protein; RNAP: RNA polymerase.

**Table 4 biomedicines-10-02469-t004:** Univariate analysis of factors predicting the development of malignant disease.

	Odds Ratio	95% CI	Standard Error	*p* Value
**Age (years)**	1.5	0.13–17.9	0.027	0.737
**Male gender**	2.5	0.72–9.0	0.32	0.147
**Smoker**	1.9	0.66–5.3	0.27	0.240
**Diagnosis with IPF**	1.4	0.58–3.5	0.23	0.440
**HRCT feature**				
Definite UIP	1.2	0.46–3.1	0.24	0.710
NSIP	0.80	0.17–3.7	0.39	0.775
NSIP with OP	2.3	0.46–12.1	0.42	0.309
**Meeting IPAF criteria**	2.2	0.78–6.1	0.26	0.136
**Positive of any antoantibody**	3.1	1.2–7.7	0.24	0.017 *
**Autoantibodies included in the criteria of IPAF**				
Antinuclear antibody				
ANA titer ≧ 1:320	3.4	1.2–9.8	0.27	0.024 *
Nucleolar pattern	2.8	0.71–11.4	0.35	0.141
Discrete speckled pattern	1.6	0.18–14.2	0.56	0.683
RF more than upper limit × 2	0.92	0.25–3.3	0.33	0.902
Anti-CCP antibody	0.67	0.82–5.5	0.54	0.709
Anti-ARS antibody	3.7	0.88–15.5	0.37	0.074
Anti-dsDNA antibody	2.3	0.11–1.7	0.35	0.220
Anti-Ro52 antibody	3.2	0.93–11.2	0.32	0.065
Anti-Ro60 antibody	8.1	0.49–134.3	0.72	0.145
**Autoantibodies not included in the criteria of IPAF**				
Anti-CENP-A antibody	2.0	0.21–18.6	0.57	0.547
Anti-CENP-B antibody	1.3	0.15–11.4	0.55	0.807
Anti-RNAP I/III antibody	8.1	0.49–134.3	0.72	0.145
Anti-Su/Ago2 antibody	2.7	0.27–26.8	0.59	0.405
Anti-Ki/SL antibody	8.1	0.49–134.3	0.72	0.145
**Serum data**				
KL-6 (IU/mL)	0.28	0.0020–37.4	0.00024	0.607
CRP (mg/dL)	0.32	0.00079–127.4	0.094	0.708
**Pulmonary function**				
FVC (%)	3.1	0.33–30.4	0.011	0.323
D_LCO_ (%)	0.42	0.037–4.8	0.010	0.480
**Developing AE**	1.1	0.37–3.1	0.27	0.907
**Developing Connective tissue disease**	4.2	0.72–24.3	0.45	0.111

CI: confidence interval; IPF: idiopathic pulmonary fibrosis; HRCT: high resolution computed tomography; IIP: idiopathic interstitial pneumonia; ILD: interstitial lung disease; NSIP: non-specific interstitial pneumonia; COP: cryptogenic organizing pneumonia; IIP: idiopathic interstitial pneumonia; UIP: usual interstitial pneumonia; IPAF: interstitial pneumonia with autoimmune features; RF: rheumatoid factor; CCP: cyclic citrullinated peptide; RNP: ribonucleoprotein; RNAP: RNA polymerase; ARS: aminoacyl tRNA synthetase; CENP: centromere protein; FVC: forced vital capacity; DLCO: diffusing capacity of the lung for carbon monoxide; AE: acute exacerbation; Age, ILD subtype, HRCT features, serum biomarker level, and pulmonary function were evaluated at the time of diagnoses with idiopathic interstitial pneumonia. * A *p*-value < 0.05 represented statistical significance.

**Table 5 biomedicines-10-02469-t005:** Multivariate analysis of factors associated with malignant disease.

**(A) Model 1**				
	**Odds Ratio**	**95% CI**	**Standard Error**	** *p* ** **Value**
**Male gender**	3.7	1.0–13.5	0.33	0.029 *
**Positive for any autoantibody**	3.9	1.5–10.1	0.24	0.004 *
**Age at the time of diagnosis with IIPs**				N.S.
**Smoker**				N.S.
**(B) Model 2**				
	**Odds Ratio**	**95% CI**	**Standard Error**	** *p* ** **Value**
**Male gender**	3.9	1.0–15.3	0.35	0.049 *
**Antinuclear antibody titer** **≧ 1:320**	4.2	1.4–13.3	0.42	0.013 *
**Anti-ARS antibody**	6.5	1.2–34.1	0.29	0.026 *
**Age at the time of diagnosis with IIPs**				N.S.
**Smoker**				N.S.
**Anti-Ro52 antibody**				N.S.

CI: confidence interval; ARS: aminoacyl tRNA synthetase; N.S.: not significant. * A *p*-value < 0.05 represented statistical significance.

## Data Availability

The data presented in this study are available on request from the corresponding author. The data are not publicly available due to ethical considerations.

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
