# Peer review of "Positive Autoantibody Is Associated with Malignancies in Patients with Idiopathic Interstitial Pneumonias"

_biomedicines, 2022, doi:10.3390/biomedicines10102469_

Round 1

Reviewer 1 Report

This is a very interesting manuscript describing the association of certain autoantibodies (antinuclear, anti-ARS, anti-Ro52) with malignant disease in a cohort of patients with idiopathic interstitial pneumonia. I only have few comments; firstly, the manuscript would benefit from language checking by native speakers, and secondly, Tables 4 and 5 should be "odds ratios" instead of "odds ratios".

The manuscript is scientifically sound and in my opinion worth publishing.    Novelty: This is, to my knowledge, the first study describing an association between autoantibodies and malignant disease in an IIP cohort. Originality: The originality is rather low as it is a standard cohort study. The results seem to be relevant for a specific audience that is interested in IIP and autoantibodies, but not for a broader audience. Scientificity: The methods are described in detail such that they are reproducible by others. Data presentation and interpretation are also scientifically sound.

References: The manuscript cites the relevant literature.

Author Response

Response to Comments

Reviewer 1

This is a very interesting manuscript describing the association of certain autoantibodies (antinuclear, anti-ARS, anti-Ro52) with malignant disease in a cohort of patients with idiopathic interstitial pneumonia. I only have few comments; firstly, the manuscript would benefit from language checking by native speakers, and secondly, Tables 4 and 5 should be "odds ratios" instead of "odds ratios".

Response

Thank you for your comments.

This manuscript was proposed to language checking by native speakers, again and revised.

For example, we have changed ‘antibody’ to ‘antibodies’ at various places throughout the text, to reflect the fact that multipl antibodies are detected. And then, there are two sections numbered '3.2'. We changed the section number in result correctly.

We changed “Odd’s ratio” to “odds ratio”

Additional revision

We changed the descriptions of “Analyses of serum autoantibodies” in Method (page 3), according to homology check with previous report.

Reviewer 2 Report

1) Background and Objective: Autoantibodies occasionally appear and are associated with clinical outcome in patients with idiopathic interstitial pneumonias (IIPs). We analyzed the association between autoantibody and having malignant disease in the present study. Methods: We retrospectively analyzed medical records and serum obtained from 193 consecutive IIPs patients at the time of their initial diagnosis. Comprehensive analyses of autoantibodies were performed using immunoprecipitation or enzyme-linked immunosorbent assay. Clinical data associated with term from IIPs diagnoses to having malignant disease were analyzed using logistic regression analysis. Variables with less than 0.10 of P value in univariate analyses and age, male gender, and smoking status as common risk factors of malignant disease were analyzed by multivariate analysis. Results: Twenty-two of 193 patients (11.4%) with IIPs had malignant disease including primary lung cancer in 9 patients (40.9%), gastric cancer in 4 (1.8%), rectal cancer in 3 (1.4%), esophageal, pharynx, breast, ovarium, breast, bladder, prostate, liver and skin in 1 (4.5%) each, respectively. In univariate analysis, positive of any autoantibodies (Odd’s ratio (OR), 3.1; confidence interval (CI),1.2-7.7; p = 0.017), more than 320 index of titer of antinuclear antibody (OR, 3.4; CI, 1.2-9.8; p = 0.024) were significantly associated with having malignant disease. Positive of anti-ARS antibody (OR, 3.7; CI, 0.88-15.5; p = 0.074) and positive of anti-Ro52 antibody (OR, 3.2; CI, 0.93-11.2; p = 0.065) were tended to be associated with that. In multivariate analysis, the independent risk factors of having malignant disease were male gender (OR, 3.7; CI, 1.0-13.5; p = 0.029) and positive of any autoantibody (OR, 3.9; CI, 1.5-10.1; p = 0.0041) in model 1 and male gender (OR, 3.9; CI, 1.0-15.3; p = 0.049), more than 320 index of antinuclear antibody titer (OR, 4.2; CI, 1.4-13.3; p = 0.013), and positive of anti-ARS antibody (OR, 6.5; CI, 1.2-34.1; p = 0.026) in model 2. Conclusion: In the present study, positive of any autoantibodies, antinuclear and anti-ARS antibodies, and male gender were independent risk factors for having cancer in IIPs patients. Presence of anti-Ro52 antibody was significantly associated with having malignant disease by univariate analysis. We showed that prediction of cancer risk is one of the clinical significances of evaluating autoantibodies in patients with IIPs. The abstract is quite rumbling, please divide the two sections of background and objective.

2) Testing autoantibodies in IIP patients may help the early diagnosis of malignancies. I would replace may help” with “might”.

3) 1. Introduction L45-49. Idiopathic interstitial pneumonias (IIPs) are diffuse inflammatory and/or fibrotic  lung diseases that exclude interstitial lung diseases (ILDs) with known causes such as connective tissue diseases (CTD) [1-3]. Patients with IIPs occasionally have clinical features suggesting the presence of autoimmune disease including positive for autoantibodies without meeting the diagnostic criteria of any CTD. Please improve this paragraph and add these references:

a- High-Resolution Computed Tomography: Lights and Shadows in Improving Care for SSc-ILD Patients. Diagnostics (Basel). 2021 Oct 22;11(11):1960. doi: 10.3390/diagnostics11111960.

b- The role of chest CT in deciphering interstitial lung involvement: systemic sclerosis versus COVID-19. Rheumatology (Oxford). 2022 Apr 11;61(4):1600-1609. doi: 10.1093/rheumatology/keab615.

c- Research Progress in the Molecular Mechanisms, Therapeutic Targets, and Drug Development of Idiopathic Pulmonary Fibrosis. Front Pharmacol. 2022 Jul 21;13:963054. doi: 10.3389/fphar.2022.963054.

4) Introduction. L98-100. We hypothesized that the presence of autoantibodies is a risk factor for malignancies in patients with IIP on the basis of these previous studies and attempted to clarify their association in the present study. Please, improve the description of study aim and underline the novelty of the study.

5) 2.1. Patients L102-113. We retrospectively analyzed the medical records and sera from 193 consecutive IIP  patients at the time of their initial diagnosis when they visited our hospital between 2012  and 2021. All patients with IIP were retrospectively diagnosed and classified by multidis-  ciplinary discussion (MDD) diagnosis using medical records, chest X-ray, and high-reso-lution computed tomography (HRCT) based on the global guidelines for IIP [1, 3]. MDD  diagnoses were performed by two respiratory physicians and three chest radiologists  and/or one chest pathologist specializing in diffuse lung disease. Classification criteria of  IPAF were based on the 2015 European Respiratory Society/American Thoracic Society Task Force research statement [4]. Acute exacerbation (AE) of ILD was defined based on  a previous report [36]. Patients diagnosed with secondary ILD such as hypersensitivity  pneumonia, CTD, or pneumoconiosis were excluded from this study. In this study, both  former and current smokers were defined as smokers. Please, insert here the citation of Table 1.

6) Table 1. Characteristics of patients and outcomes. Please add the most important statistical values.

7) Table 2. Prevalence of autoantibodies. Please add the most important r values.

8) 4. Discussion L261-262 In the present study, we demonstrated that positive for any autoantibodies and male were independent risk factors for malignancies in patients with IIPs. I would replace that positive for any autoantibodies” with “that any autoantibodies positivity”.

9) 5. Conclusions L318-322. In the present study, the presence of any autoantibodies, antinuclear and anti-ARS  antibodies, and male, were independent risk factors for cancer in IIPs patients. The presence of anti-Ro52 antibody tended to be associated with malignant disease by univariate  analysis. Testing for autoantibodies may help the early diagnosis of malignant disease in  patients with IIPs. Please, underline the novelty of the study and the clinical implication of the study.

Author Response

Response to Comments

Reviewer 2

Comment 1

The abstract is quite rumbling, please divide the two sections of background and objective.

Response

I am sorry that we accidentally pasted the pre-revised version of the abstract in the text box for the abstract. We have provided a more simplified abstract as follows in our uploaded WORD and PDF manuscripts.

We have asked the editorial to change the abstract in the text box to the simplified version below.

Abstract: Various autoantibodies are associated with clinical outcomes in patients with idiopathic interstitial pneumonias (IIP). We analyzed the association between autoantibody and malignancies in IIP patients. Comprehensive analyses of autoantibodies were retrospectively performed using immunoprecipitation and enzyme-linked immunosorbent assay in 193 consecutive IIP patients. Cancer-related factors were analyzed using logistic regression analysis. Twenty-two of 193 patients (11.4%) with IIP had malignant disease. In univariate analysis, positive for any autoantibody (odds ratio (OR), 3.1; confidence interval (CI), 1.2–7.7; p=0.0170) and antinuclear antibody titer≧1:320 (OR, 3.4; CI, 1.2–9.8; p=0.024) were significantly associated with malignancies. Positive anti-ARS (aminoacyl tRNA synthetase, OR, 3.7; CI, 0.88–15.5; p=0.074) and anti-Ro52 antibody (OR, 3.2; CI, 0.93–11.2; p=0.0645) tended to be associated with malignancies. In multivariate analysis, independent risk factors were male (OR, 3.7; CI, 1.0-13.5; p=0.029) and positive for any autoantibody (OR, 3.9; CI, 1.5–10.1; p=0.004) in model 1 and male (OR, 3.9; CI, 1.0–15.3; p=0.0498), antinuclear antibody titer≧1:320 (OR, 4.2; CI, 1.4–13.3; p=0.0125), and positive for anti-ARS antibody (OR, 6.5; CI, 1.2–34.1; p=0.0263) in model 2. Positive for any autoantibody, antinuclear and anti-ARS anti-bodies, and male were independent risk factors for malignancies in IIP patients. Testing autoantibodies in IIP patients may help the early diagnosis of malignancies.

Comment 2

Page 1 (abstract) and Page 11 (Conclusion)

Testing autoantibodies in IIP patients may help the early diagnosis of malignancies.

I would replace “may help” with “might”.

Response

Thank you for your comment.

We revised the manuscript as reviewer pointed.

Comment 3

  1. Introduction L45-49. Idiopathic interstitial pneumonias (IIPs) are diffuse inflammatory and/or fibrotic lung diseases that exclude interstitial lung diseases (ILDs) with known causes such as connective tissue diseases (CTD) [1-3]. Patients with IIPs occasionally have clinical features suggesting the presence of autoimmune disease including positive for autoantibodies without meeting the diagnostic criteria of any CTD.

Response

Thank you for your comment. References 4, 5, 6 have been added and the paragraphs have been improved as follows. We change the reference number after No. 6.

Idiopathic interstitial pneumonias (IIPs) are diffuse inflammatory and/or fibrotic lung diseases that exclude interstitial lung diseases (ILDs) with known causes such as connective tissue diseases (CTD) [1-3]. Evaluation of chest CT supports early diagnosis of ILD [4,5]. Recent studies have identified the key role of crosstalk between dysregulated epithelial cells, mesenchymal, immune, endothelial cells, genetic mutations and environ-mental factors (e.g., smoking) in idiopathic pulmonary fibrosis, a subtype of IIPs [6]. Patients with IIPs occasionally have clinical features suggesting the presence of autoimmune disease including positive for autoantibodies without meeting the diagnostic criteria of any CTD. For example, systemic sclerosis in early or mild disease, disease specific symp-toms often do not appear [5]. Interstitial pneumonia with autoimmune features (IPAF) is a term that represents a subset of patients with this phenotype [7].

Comment 4

Introduction. L98-100. We hypothesized that the presence of autoantibodies is a risk factor for malignancies in patients with IIP on the basis of these previous studies and attempted to clarify their association in the present study. Please, improve the description of study aim and underline the novelty of the study.

Response

Thank you for your comment.

We revised introduction as follows.

We attempted to clarify that the presence of autoantibodies is associated with malignancies in patients with IIP on the basis of these previous studies. This is the first study to analyze the association between autoantibodies and malignancies in IIP.

Comment 5

Patients L102-113. Please, insert here the citation of Table 1.

Response

Thank you for your comment.

We insert the citation of Table 1.

Comment 6

Table 1. Characteristics of patients and outcomes. Please add the most important statistical values

Comment 7

Table 2. Prevalence of autoantibodies. Please add the most important r values.

Response

Thank you for your comment.

We added r values in table 1 and 2.

We added the following description in Statistical analyses.

Correlations between the two parameters were evaluated using Spearman's rank correlation coefficient.

Comment 8

Discussion L261-262 In the present study, we demonstrated that positive for any autoantibodies and male were independent risk factors for malignancies in patients with IIPs. I would replace “that positive for any autoantibodies” with “that any autoantibodies positivity”.

Response

Thank you for your comment.

We revised the Discussion as the reviewer suggested.

Comment 9

9) 5. Conclusions L318-322. In the present study, the presence of any autoantibodies, antinuclear and anti-ARS  antibodies, and male, were independent risk factors for cancer in IIPs patients. The presence of anti-Ro52 antibody tended to be associated with malignant disease by univariate  analysis. Testing for autoantibodies may help the early diagnosis of malignant disease in  patients with IIPs. Please, underline the novelty of the study and the clinical implication of the study.

Response

Thank you for your comment.

We underline the novelty of the study and the clinical implication of the study.

In the present study, the presence of any autoantibodies, antinuclear and anti-ARS antibodies, and male, were independent risk factors for cancer in IIPs patients. The presence of anti-Ro52 antibody tended to be associated with malignant disease by univariate analysis. Testing for autoantibodies might help the early diagnosis of malignant disease in patients with IIPs.

Additional revision

We changed the descriptions of “Analyses of serum autoantibodies” in Method (page 3), according to homology check with previous report.

Reviewer 3 Report

In my opinion, the work is extremely useful because it talks about new diagnostic possibilities and helps in defining and identifying early risk factors for malignant disease in patients with idiopathic interstitial pneumonias. The introduction, methods, results and discussion as well as the references are well written. Only one minor correction is needed: the authors should write uniformly the p value, ie., with 3 or 4 decimals, especially in tables. So please decide.

Author Response

Response to Comments

Reviewer 3

Comment

the authors should write uniformly the p value, ie., with 3 or 4 decimals, especially in tables.

Response

Thank you for your comment.

We standardized the p value to 3 decimals.

Additional revision

We changed the descriptions of “Analyses of serum autoantibodies” in Method (page 3), according to homology check with previous report.